# Periglacial Lake Origin Influences the Likelihood of Lake Drainage in Northern Alaska

**Mark Jason Lara** [1,2,*] and **Melissa Lynn Chipman** [3]

1  Department of Plant Biology, University of Illinois, Urbana, IL 61801, USA
2  Department of Geography, University of Illinois, Urbana, IL 61801, USA
3  Department of Earth and Environmental Sciences, Syracuse University, Syracuse, NY 13244, USA; mlchipma@syr.edu
*  Correspondence: mjlara@illinois.edu

**Abstract:** Nearly 25% of all lakes on earth are located at high latitudes. These lakes are formed by a combination of thermokarst, glacial, and geological processes. Evidence suggests that the origin of periglacial lake formation may be an important factor controlling the likelihood of lakes to drain. However, geospatial data regarding the spatial distribution of these dominant Arctic and subarctic lakes are limited or do not exist. Here, we use lake-specific morphological properties using the Arctic Digital Elevation Model (DEM) and Landsat imagery to develop a Thermokarst lake Settlement Index (TSI), which was used in combination with available geospatial datasets of glacier history and yedoma permafrost extent to classify Arctic and subarctic lakes into Thermokarst (non-yedoma), Yedoma, Glacial, and Maar lakes, respectively. This lake origin dataset was used to evaluate the influence of lake origin on drainage between 1985 and 2019 in northern Alaska. The lake origin map and lake drainage datasets were synthesized using five-year seamless Landsat ETM+ and OLI image composites. Nearly 35,000 lakes and their properties were characterized from Landsat mosaics using an object-based image analysis. Results indicate that the pattern of lake drainage varied by lake origin, and the proportion of lakes that completely drained (i.e., >60% area loss) between 1985 and 2019 in Thermokarst (non-yedoma), Yedoma, Glacial, and Maar lakes were 12.1, 9.5, 8.7, and 0.0%, respectively. The lakes most vulnerable to draining were small thermokarst (non-yedoma) lakes (12.7%) and large yedoma lakes (12.5%), while the most resilient were large and medium-sized glacial lakes (4.9 and 4.1%) and Maar lakes (0.0%). This analysis provides a simple remote sensing approach to estimate the spatial distribution of dominant lake origins across variable physiography and surficial geology, useful for discriminating between vulnerable versus resilient Arctic and subarctic lakes that are likely to change in warmer and wetter climates.

**Keywords:** permafrost thaw; thermokarst; yedoma; Landsat; Alaska; thaw settlement; glaciation

## 1. Introduction

Quaternary glaciations have left a large footprint on the global freshwater system. The highest densities of freshwater lakes are located at high latitudes (e.g., ~50 and 75°N), but particularly concentrated within the limits of the Last Glacial Maximum in Canada, Scandinavia, Russia, and Alaska [1,2]. Northern lakes have a variety of origins, including thermokarst (i.e., surface subsidence via ground ice melt), glacial activity that results in depressions (e.g., kettles, cirques) and dams (e.g., ice-dammed and moraine-dammed lakes), and/or hydrogeological processes (i.e., fluvial, floodplain, and coastal erosion) that shape and reshape land surfaces over millennial timescales. Lakes have become synonymous with permafrost regions, yet lake abundance and origin vary across space with permafrost conditions (i.e., ice-rich and ice-poor) and landscape history.

Lakes in northern Alaska have four principal lake origins: (i) non-yedoma thermokarst lakes, (ii) yedoma thermokarst lakes, (iii) glacial lakes, and (iv) volcanic origin lakes [3]. Thermokarst lakes (non-yedoma) form in closed depressions by the thaw and collapse of

ice-rich permafrost or massive ground ice, which can develop shallow thaw settlements of ~1 to 3 m in depth [4–6]. Similar thermokarst processes occur in extremely ice-rich and silt-dominated yedoma deposits, which develop deep ~10 to 20 m thaw settlements [6–8]. These deep thaw settlements are also characteristic of glacial lakes (i.e., kettle lakes) that form following the collapse of stagnant buried ice blocks [9]. Both yedoma and glacial lake boundaries are usually circular in shape as the melting of high volumetric ground ice (70–90%) makes them round over time [7,8,10]. Although there are many types of lakes that exist in Arctic permafrost regions (i.e., tarns, bedrock basin lakes, glacier-carved lakes, inter-dune lakes, oxbows, and lagoon lakes [11]), the most commonly found lake types in lowland and upland shrub-tussock tundra environments include thermokarst (non-yedoma), yedoma, and glacial lakes (primarily kettle lakes). However, existing geospatial datasets capable of characterizing the spatial distribution of these common glacial and periglacial lake types over large spatial gradients are extremely limited [11].

Recent evidence suggests that periglacial lake origin may be a strong control on: (i) biophysical properties [12,13], (ii) biogeochemical properties [14–16]), and (iii) the likelihood of periglacial lakes to drain [17,18]. Therefore, knowledge of the origin and spatial distribution of glacial and periglacial lakes are key to improve our understanding of the implications of climate change on landscape-level functioning. Here, we develop a remote sensing approach to map the distribution of thermokarst (non-yedoma), yedoma, glacial, and volcanic lakes using the Arctic Digital Elevation Model (DEM), Landsat imagery, and ancillary data products. We develop a Thermokarst lake Settlement Index (TSI) to detect key thermokarst and non-thermokarst lake properties, while leveraging existing geospatial data products to create a lake origin map of the northern tundra region of Alaska. Furthermore, these geospatial data products are used with decadal time-scale lake drainage analyses to test the hypothesis that lake origin influences the likelihood of northern periglacial lakes to drain [14,17].

## 2. Materials and Methods

### 2.1. Study Region

The study domain encompasses several Arctic and subarctic ecoregions of northern Alaska (Figure 1). Ecoregions include Arctic tundra (Brooks Foothills, Brooks Range), Bering tundra (Kotzebue Sound Lowlands, Seward Peninsula), and Intermontane boreal (Davidson Mountains, Kobuk Ridges and Valleys) [19]. We consider a diverse assemblage of lake types primarily situated in shrub-tussock tundra [20], which span a broad range of glacial history, substrates, topography, and climate (Figure 1). Although continuous permafrost (>90% of the landscape underlain by permafrost) dominates our study domain, discontinuous permafrost (50–90% of the landscape underlain by permafrost) may be found on the southern margin of the Seward Peninsula [21]. Soil substrates are highly heterogeneous, including silty, sandy, colluvial, peat, and glaciated ecological landscapes [21–23]. The topography across this region is highly variable, with rugged mountain peaks in the Brooks Range (elevation mean $\pm$ stdev: $892 \pm 415$ m.a.s.l.), upland terrain in the Brooks Foothills and Davidson Mountains (elevation: $308 \pm 210$ and $632 \pm 183$ m.a.s.l., respectively), and lowland areas and floodplains in the Kobuk Ridges and Valleys, Kotzebue Sound Lowlands, and the Seward Peninsula (elevation: $215 \pm 163$, $24 \pm 18$, and $194 \pm 155$ m.a.s.l., respectively). The climate varies across the study domain as mean annual air temperature ranges from $-1.9$ °C to $-10.5$ °C and precipitation from 190 to 719 mm (CRU, www.cru.uea.ac.uk/, accessed on 16 April 2019). Annual temperature and precipitation also vary across ecoregions: Seward Peninsula ($-3.5 \pm 0.6$, $417 \pm 82$), Kobuk Ridges and Valleys ($-3.4 \pm 0.8$, $424 \pm 61$), Kotzebue Sound Lowlands ($-4.1 \pm 0.3$, $351 \pm 37$), Davidson Mountains ($-5.1 \pm 1.0$, $252 \pm 43$), Brooks Range ($-6.3 \pm 1.6$, $476 \pm 108$), and the Brooks Foothills ($-8.8 \pm 1.3$, $344 \pm 79$).

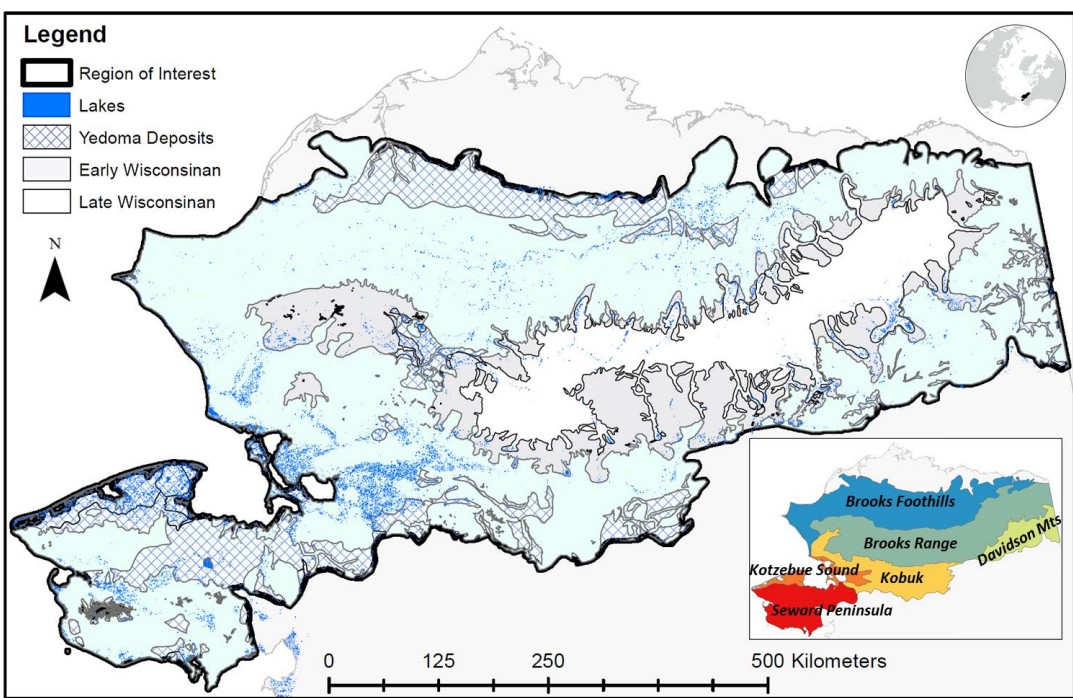

**Figure 1.** Upland tussock-shrub tundra region of Alaska. The bolded line bounds our region of interest, which includes (i) nearly 35,000 lakes greater than 1 ha in size, (ii) the known yedoma deposits, (iii) the Wisconsinan glacial extent during the last glacial maximum, and (iv) seven ecoregions: Brooks Foothills, Brooks Range, Davidson Mountains, Kobuk Ridges and Valleys, Kotzebue Sound Lowlands, and the Seward Peninsula.

We focus on the four principal lake origins: (i) non-yedoma thermokarst lakes (henceforth Thermokarst lakes), (ii) yedoma thermokarst lakes (henceforth Yedoma lakes), (iii) Glacial lakes, and (iv) volcanic lakes (henceforth Maar lakes; Table 1). Thermokarst lakes are widespread in lowland (low topographic relief) ice-rich permafrost terrain of Alaska. Although thermokarst lakes may be large, shallow, and elliptical on the Arctic Coastal Plain of northern Alaska [10], thermokarst lake morphology varies widely in terms of wind direction, differences in lake bluff (i.e., bank) height, and permafrost conditions across upland tussock-shrub tundra ranging from the Seward Peninsula to the Foothills of the Brooks Range [24,25]. Extremely ice-rich syngenetic yedoma deposits have accumulated over thousands of years in unglaciated fine silt and sand during the last glacial maximum (i.e., late Wisconsinan extent) in northern Alaska [26–29]. These organic-rich silt (loess) soils typically occur where sedimentation of fine material has been relatively continuous such as deltas, flood plains, and loess belts [30], and are commonly found in alluvial plains, hillslopes, and polygonal lowlands across Alaska [31,32]. Known yedoma deposits in Alaska are located (i) near the southern boundary of the Arctic Coastal Plain, (ii) within the Noatak National Preserve (located in the northwestern Kobuk Ridges and Valleys and the western Brooks Range), and (iii) throughout the Seward Peninsula (Figure 1). Glacial lakes are found throughout relatively ice-poor deglaciated terrain (e.g., late Wisconsinan extent), ranging in size between 5 m and 13 km in diameter and 45 m in depth. These post-glacial landscapes are marked by topographic features from glacial retreat, which contain isolated deposits of remnant glacial ice. Maar lakes form within a volcanic crater that develops when hot lava encroaches into the groundwater, causing a violent phreatomagmatic eruption. In the presence of permafrost, the strength of these eruptions are likely magnified, and Maar lakes on the Bering Land Bridge National Preserve are among the largest on the planet [33], ranging in size from 4 to 8 km in diameter and 30 m in depth.

**Table 1.** Description of the four principal lake origins, found across the permafrost region in northern Alaska.

| Lake Origin | Description |
|---|---|
| Thermokarst | Thermokarst lakes (non-yedoma) form in closed depressions by the thaw and collapse of ice-rich permafrost or massive ground ice; widespread in low topographic relief permafrost terrain. |
| Yedoma | Yedoma thermokarst lakes form in extremely ice-rich organic rich silt (loess) soils, where sedimentation of fine material has been relatively continuous such as deltas, flood plains, and loess belts; commonly found in alluvial plains, hillslopes, and polygonal lowlands across Alaska. |
| Glacial | Glacial lakes form following the collapse of stagnant buried ice blocks deposited following glacial recession; found throughout the late Wisconsinan glacial extent in relatively ice-poor terrain. |
| Maar | Maar lakes form within a volcanic craters that develops when hot lava encroaches into the groundwater causing a violent phreatomagmatic eruption; found in the Seward Peninsula within ice-poor terrain. |

### 2.2. Image Processing

We mapped the spatial distribution of glacial and periglacial lakes and their origins using the following platforms: Google Earth Engine (GEE), eCognition v.9.1, R v.3.6.2, and ArcGIS v.10.8.1 platforms (Figure 2). Landsat 7 ETM+ surface reflectance data were pre-processed by the United States Geological Survey and downloaded by GEE in a radiometrically, atmospherically, and geometrically terrain-corrected state. The earliest possible time period was selected to represent the lake distribution and extent prior to recent widespread lake drainage [17,18,34–36]. Therefore, we acquired 273 Landsat scenes during the ice-free period (1 July to 1 September) between 1985 and 1989 to create a seamless image mosaic. The mosaic dataset included the following multispectral bands: blue (0.45–0.52 μm), green (0.52–0.60 μm), red (0.63–0.69 μm), near-infrared (0.77–0.90 μm), short-wave infrared 1 (1.55–1.75 μm), and short-wave infrared 2 (2.08–2.35 μm). The Fmask algorithm and short-wave infrared bands were used to filter clouds across temporally dense image stacks [37]. Median pixel values across cloud masked image stacks were selected to generate the image composite used to extract surface water (Figure 2).

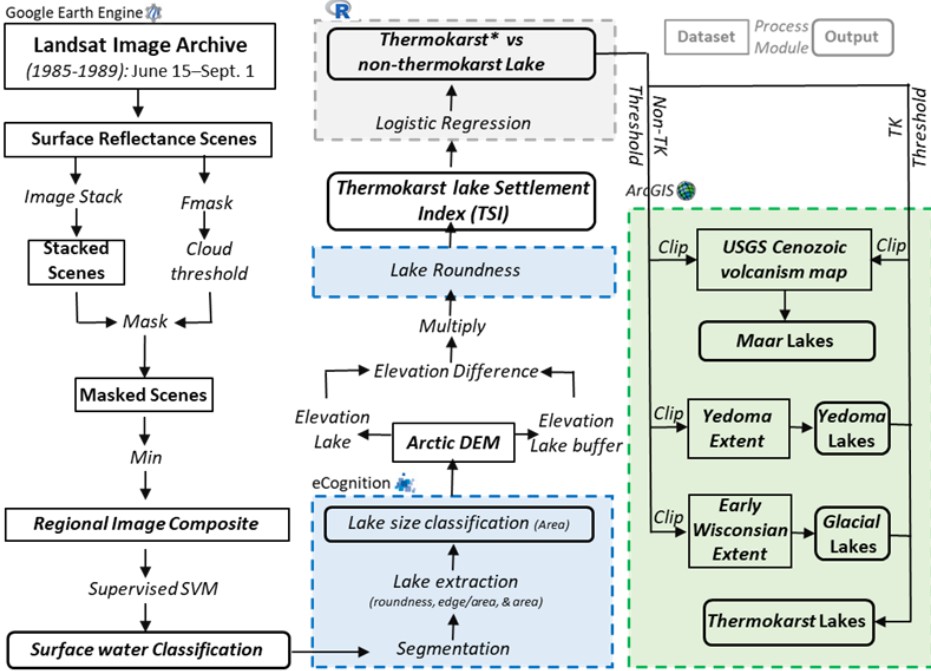

**Figure 2.** Schematic representation of the data processing workflow used to derive the Thermokarst lake Settlement Index (TSI) and map the distribution of lake origin across our study domain. Processing within Google Earth Engine (GEE), eCognition, R, and ArcGIS are indicated by background colors white, blue, grey, and green, respectively.

### 2.3. Surface Water Classification

Waterbodies were extracted within GEE using a supervised support vector machine (SVM) classifier spanning our study domain (Figure 1). We selected 200 training sites that represented the full spectral range of surface waterbodies such as ponds, rivers, lakes, and coastal water. The surface water classes were exported into the Object-Based Image Analysis (OBIA) software eCognition, where ponds, rivers, and coastal water were identified and masked using spectral and morphological (i.e., roundness) metrics [38,39]. Similar to [39], lakes were differentiated from ponds using the surface area thresholds (i.e., ponds < 1 ha), resulting in 32,690 lakes considered in this analysis. Lake area, shoreline length (i.e., perimeter), and the shoreline development ratio (SDR) were computed using the equation $L/(2\sqrt{\prod A})$, where L is the length of the shoreline (meters) and A is the area of the lake (m$^2$). The SDR may be interpreted as a measure of lake shape irregularity as compared to the shoreline length of a perfectly circular lake of equal area. The final lake distribution map was inspected for classification errors associated with hill shadows and manually removed (<0.01% of all lakes were edited) and imported into GEE.

### 2.4. Decadal Lake Drainage

To estimate the spatial and temporal patterns of "complete" lake drainage, we used the 1985–1989 lake extent to extract lake area from a 2015–2019 Landsat image mosaic processed in the same manner as described above. A total of 1186 Landsat scenes were acquired during the ice-free period (1 July to 1 September), between 2015 and 2019. Lake drainage was computed as the difference in the 1985–1989 lake extent from that estimated during the 2015–2019 time period. Here, we refer to lake drainage as "complete" drainage rather than catastrophic drainage, as catastrophic drainage occurs abruptly in time [11], while complete drainage is irrespective of time. Although the disappearance of lakes (i.e., 100% decrease in the initial lake extent) may occur following drainage, ponds and small lakes typically form within the initial lake boundary [6,40]. Thus, we define "complete" lake drainage as a lake area reduction >60% of the initial (i.e., 1985–1989) lake area.

### 2.5. Lake Origin Classification

Due to the known morphological differences between lakes of varying origin [6–8,10,33], we used both two-dimensional (i.e., lake roundness) and three-dimensional (i.e., lake settlement) morphometrics to generate a Thermokarst lake Settlement Index (TSI). While lake roundness was computed by the OBIA, lake settlement was measured as the difference in elevation (m.a.s.l.) between the adjacent lake shoreline bluff and the lake surface water using the 2 m spatial resolution Arctic Digital Elevation Model (DEM). The TSI is a scalar function described as follows:

$$TSI = (Elev_S - Elev_B) \times Roundness$$

where the elevation difference between the lake surface elevation (Elev$_S$) and the lake bluff elevation (Elev$_B$) were computed as the median elevation at the lake surface and the median elevation within a 500 m buffer adjacent to each lake (Figure A1). This elevation difference was multiplied by lake roundness (ranging from 0 to 1) to produce a continuous index ranging from negative (i.e., thermokarst lakes) to positive (i.e., non-thermokarst lakes) values.

To develop a discrete map of periglacial lake origin, we quantitatively identified a threshold to classify the continuous TSI into binary classes of thermokarst and non-thermokarst lakes. Logistic regression was used to determine the TSI threshold at which the probability of thermokarst and non-thermokarst lakes was >50%. Sub-meter high spatial resolution imagery and ground-based observations (e.g., [6,21–23,41]) were used to identify lake regions with a high proportion of thermokarst and non-thermokarst lakes, respectively (Figure A2). Data from these lake regions were used to implement the logistic regression analyses. These lake regions included (i) 3335 thermokarst (non-yedoma) lakes located in the southwestern Noatak National Preserve and the Kobuk Delta covering 3738 km$^2$

and (ii) 1854 "non-thermokarst" lakes (includes yedoma and glacial origin lakes) located in the northern Seward Peninsula, north-eastern Noatak National Preserve, and on the foothills of the Brooks Range, covering 6610 km$^2$. We evaluated the performance of the logistic regression analysis using McFaddens's $R^2$ value, contingency tables, and the area under the receiving operation characteristic curve (AUROC). The McFaddens's $R^2$ value ranges from 0.0 to 1.0, where the closer to 1, the greater the predictive power. Contingency tables are used to determine the overall accuracy and kappa coefficient of reference versus mapped categories (i.e., thermokarst versus non-thermokarst lakes). The AUROC ranges from 0.5 to 1.0, where values above 0.8 indicate that the model well discriminates between two categories.

All lakes were initially classified as either thermokarst or non-thermokarst lakes using the TSI threshold. A United States Geological Survey map of late Cenozoic volcanic centers was used to directly identify lakes of volcanic origin in northern Alaska [42]. Similarly, we used the database of Ice-rich Yedoma Permafrost [43] and the Pleistocene Maximum Wisconsinan glacier extent [29] to determine the likely spatial location of Yedoma and Glacial lakes. However, both yedoma permafrost and Wisconsian extents are generalized polygons that encompass a wide range of heterogeneous terrain and lakes. Therefore, we used the TSI threshold to extract low-relief thermokarst (non-yedoma) lakes from both yedoma and Wiscosinan spatial extents using the >50% TSI threshold. All remaining lakes within these respective spatial extents were classified as either Yedoma or Glacial lakes.

## 3. Results

We estimated lake-specific morphological metrics across 32,690 lakes spanning 389,259 km$^2$ of northern tundra in Alaska (Figure 1). The majority of the lakes (56.4%) were located at low elevations (<56 m. a.s.l.) within coastlines, river deltas, and river basins (Figure 3). Lake roundness was normally distributed across all lakes, ranging from irregular lake perimeters (low roundness) to nearly symmetrical circles (high roundness). We integrated lake roundness into the TSI to improve the discrimination between thermokarst and non-thermokarst lakes, as circular lakes (i.e., high roundness) are typically associated with non-thermokarst lakes [7,8,10]. The difference in topographical relief between lake level and the adjacent terrain elevation associated with lake settlement was generally small (~1 to 2 m), but this difference was as large as >20 m for some lakes (Figure 3).

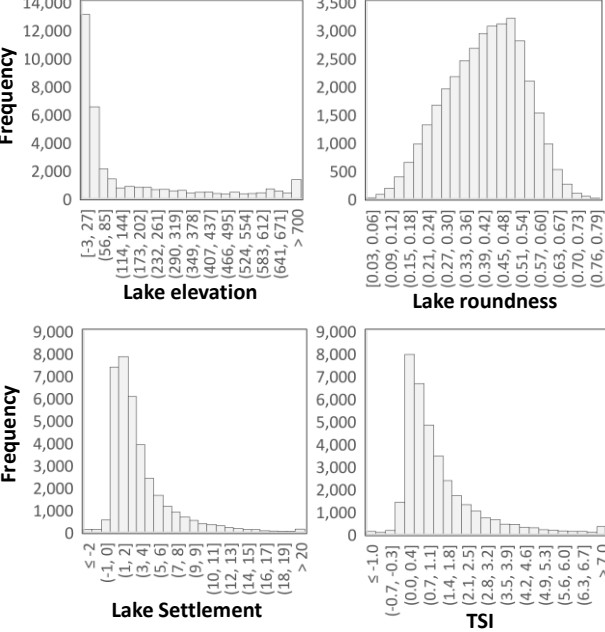

**Figure 3.** Morphological lake metrics summarized across all lakes in our northern Alaska study domain. TSI = Thermokarst lake Settlement Index.

The TSI captured the spatial heterogeneity of thermokarst and non-thermokarst lakes spanning lowland to upland gradients (Figure 4). Specifically, the spatial distribution of thermokarst lakes is well-represented along topographic gradients on the Kobuk River Delta, southwestern Noatak National Preserve, and the central Seward Peninsula (Figure 4). As anticipated, thermokarst lakes (low TSI) were commonly found in lowland terrain, whereas non-thermokarst lakes (high TSI) were increasingly identified in upland terrain with high rugosity and topographic relief (characteristic, e.g., of kettle lakes). However, there were many exceptions to this general trend, as the spatial distribution of yedoma deposits and past glacial extents included thermokarst lakes in low-relief upland terrain and non-thermokarst lakes in lowland terrain (Figure 4).

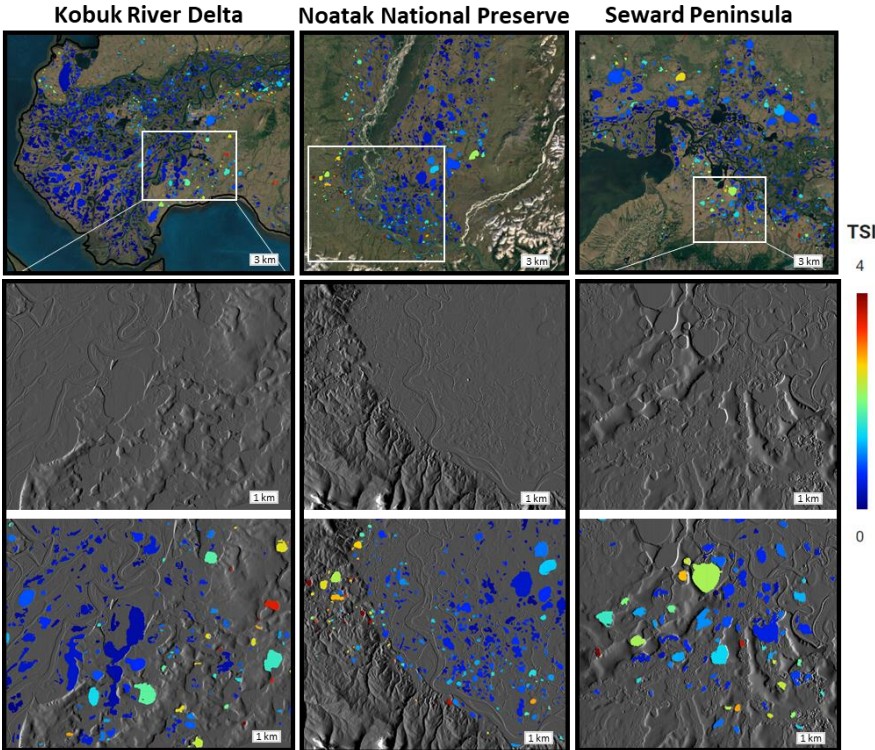

**Figure 4.** Examples of the Thermokarst lake Settlement Index (TSI) projected on Landsat scenes (**top row**) within the Kobuk River Delta (**left column**), southwestern Noatak National Preserve (**middle column**), and central Seward Peninsula (**right column**). The TSI is overlaid on a greyscale hillshade surface computed with the Arctic Digital Elevation Model (DEM) (**bottom row**).

The statistical distribution of TSI values among thermokarst and non-thermokarst lakes was markedly different (Figure 5). Index values were consistently lower in thermokarst than non-thermokarst lakes, where the median TSI values for thermokarst and non-thermokarst lakes were 0.467 and 2.657, respectively (Figure 5A). These substantial differences resulted in a high discriminatory power from the logistic regression model. All performance metrics indicated that the model well discriminated between these two lake origin categories, as McFadden's $R^2$ was 0.45, AUROC was 0.88 and the overall accuracy and kappa coefficient were 0.86 and 0.68, respectively. Therefore, the model estimated a TSI threshold of 1.27, representing a minimum probability of 50% for discriminating between thermokarst and non-thermokarst lakes (Figure 5B).

The TSI index, logistic regression thresholds, and geospatial datasets were used to generate a lake origin map of the study domain (Figure 6). This map estimated Thermokarst, Yedoma, Glacial, and Maar lakes to represent 65.5, 7.7, 26.7, and <0.1%, respectively, of all lakes in our study region. The mean lake area and perimeter for all lakes were 10.4 ha and 1462.6 m. Lake area was similar between Thermokarst, Yedoma, and Glacial lakes

(Table 2) but smallest in Yedoma lakes (mean ± sterr: 9.4 ha ± 0.8, *n* = 2543) and largest in Maar lakes (655.9 ha ± 250.8, *n* = 4), though sample size was low and variability was high for Maar lakes (Table 2). The SDR was highest (most asymmetrical) in Thermokarst lakes and lowest (most symmetrical) in Glacial lakes (Table 2). Similarly, lake roundness was lowest in Thermokarst lakes (most asymmetrical), but greater in all other lake origin classes. Thaw lake settlement progressively increased among Thermokarst (1.39 ± 0.06 m; *n* = 21,416), Yedoma (6.20 ± 0.07 m; *n* = 2543), Glacial (6.44 ± 0.04 m, *n* = 8730), and Maar (11.25 ± 4.30 m, *n* = 4) lakes, which were similar to patterns identified in the TSI (Table 2).

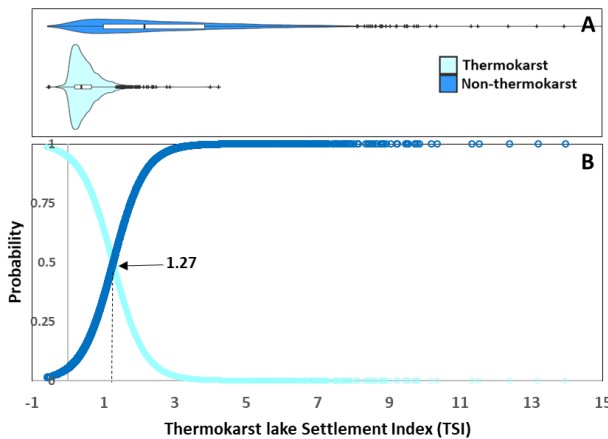

**Figure 5.** The probability of thermokarst and non-thermokarst lakes using the Thermokarst lake Settlement Index (TSI). Violin plots display the differing distribution of TSI values for 3335 thermokarst and 1854 non-thermokarst lakes, where the median TSIs were 0.467 and 2.657, respectively (**A**). Logistic regression identified a TSI of 1.27 as the 50% probability threshold to classify thermokarst and non-thermokarst lakes (**B**).

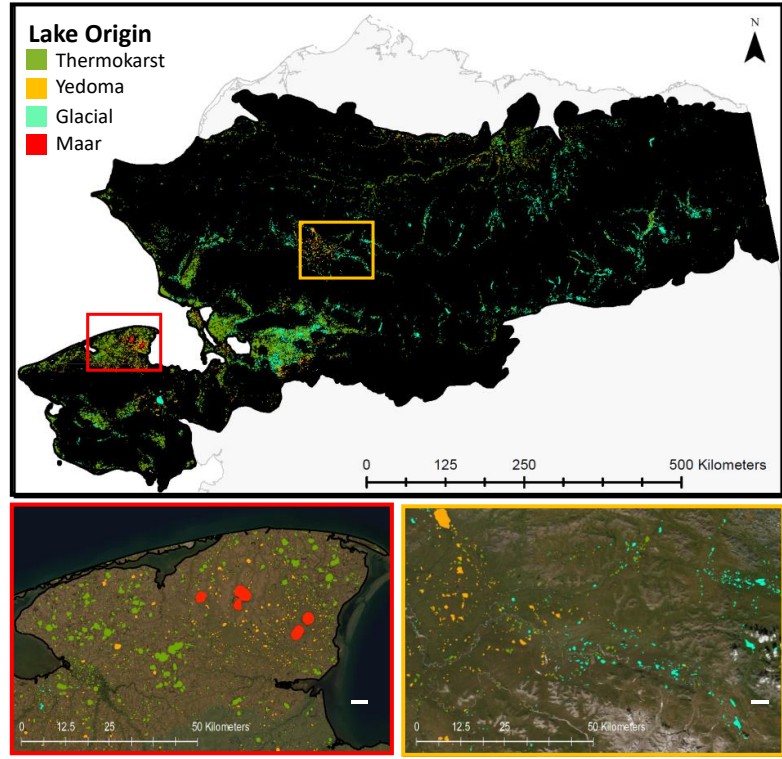

**Figure 6.** Lake origin map created for the northern tussock-shrub tundra region of Alaska. Red and orange panels show the diversity of lake origins within the northern Seward Peninsula and the eastern Noatak National Preserve, respectively.

**Table 2.** Summary of lake morphological metrics and lake drainage corresponding to lake origin (Thermokarst, Yedoma, Glacial, and Maar lakes) spanning all lakes in our northern Alaska spatial domain.

| Lake Origin | Lake Size Class | Sample Size | Lake Area (ha) | Lake Perimeter (m) | Shoreline Development Ratio [β] | Lake Roundness [β] | Lake Settlement (m) | Thermokarst Lake Settlement Index [β] | Proportion of Lakes to Completely Drain [δ] |
|---|---|---|---|---|---|---|---|---|---|
| **Thermokarst** | Large | 301 | 212.8 | 10,443.2 | 2.121 | 0.287 | 1.800 | 0.453 | 6.3% |
| | Medium | 1695 | 38.3 | 4217.3 | 1.951 | 0.309 | 1.791 | 0.514 | 6.4% |
| | Small | 19,417 | 4.5 | 1167.4 | 1.647 | 0.402 | 1.345 | 0.506 | 12.7% |
| | *All* | *21,413* | *10.1* | *1539.2* | *1.678* | *0.393* | *1.387* | *0.506* | *12.1%* |
| **Yedoma** | Large | 24 | 262.8 | 8907.7 | 1.667 | 0.381 | 7.165 | 2.602 | 12.5% |
| | Medium | 196 | 38.4 | 3393.7 | 1.581 | 0.421 | 6.998 | 2.879 | 7.7% |
| | Small | 2323 | 4.4 | 1023.0 | 1.480 | 0.475 | 6.127 | 2.832 | 9.6% |
| | *All* | *2543* | *9.4* | *1280.1* | *1.489* | *0.470* | *6.204* | *2.833* | *9.5%* |
| **Glacial** | Large | 102 | 399.1 | 12,177.0 | 1.913 | 0.311 | 11.674 | 3.117 | 4.9% |
| | Medium | 532 | 36.8 | 3497.6 | 1.653 | 0.392 | 7.629 | 2.779 | 4.1% |
| | Small | 8096 | 4.3 | 1040.2 | 1.502 | 0.464 | 6.299 | 2.791 | 9.0% |
| | *All* | *8730* | *10.9* | *1320.1* | *1.516* | *0.458* | *6.443* | *2.794* | *8.7%* |
| **Maar** | Large | 4 | 1209.0 | 18,079.3 | 1.456 | 0.498 | 19.727 | 9.884 | 0.0% |
| | *All* | *4* | *1209.0* | *18,079.3* | *1.456* | *0.498* | *19.727* | *9.884* | *0.0%* |
| **Total** | | 32,690 | 10.4 | 1462.6 | 1.620 | 0.416 | 3.114 | 1.299 | 11.0% |

[β]: unitless index; [δ]: lakes drained >60% of their initial 1985–1989 extent.

Complete lake drainage (i.e., >60% lake water loss) between 1985–1989 and 2015–2019 across our study region varied by lake origin and size class. The proportion of lakes that completely drained, ranging from highest to lowest, were Thermokarst (12.1%), Yedoma (9.5%), Glacial (8.7%), and Maar (0%) lakes, respectively (Table 2). Thermokarst lake drainage decreased by size class, as the proportions of large, medium, and small lakes that completely drained were 6.3, 6.4, and 12.7%, respectively (Table 2). Yedoma lake drainage varied by size class, but the highest and lowest proportion of lakes that drained occurred in large (12.5%) and medium (7.7%) sized lakes. With the exception of Maar lakes, which did not drain during our observation period, the most stable lakes were large and medium-sized Glacial lakes, which impacted only 4.9% (*n* = 102) and 4.1% (*n* = 532) of each respective size class. However, the prevalence of lake drainage in small glacial lakes (9.0%) was similar to that found in Thermokarst and Yedoma lakes (Table 2).

## 4. Discussion

We describe a simple geospatial protocol that leverages the Arctic DEM and Landsat data to differentiate between the origins of lake formation across permafrost ecosystems. Using the developed lake origin product and decadal patterns of lake drainage in northern Alaska, we find the most vulnerable periglacial lakes to drain between 1985–1989 and 2015–2019 were small Thermokarst (non-yedoma) and large Yedoma lakes, respectively, whereas the most resilient lakes were large and medium-sized glacial and large Maar lakes (Table 2). The discrepancy between lake drainage is likely due to the associated differences in permafrost ice content [14,17], as low to moderate ice content found adjacent to glacial and Maar lakes makes them relatively insensitive to thermo-erosional gully formation that is often responsible for abrupt lake drainage [18,35]. Although we did not observe any drainage of Maar lakes during our ~30 year observation period (likely associated with low sample size and geologic origin), they are not entirely resistant to drainage. For example, a nearby Maar lake (66°27′59″N, 164°33′13″W) observed north of Devil's Mountain on the Seward Peninsula drained nearly 150 years B.P. [44]. Collectively this evidence supports the hypothesis that lake origin and landscape history are important factors controlling the likelihood of periglacial lakes to drain.

Numerous studies have identified heterogeneous spatial and temporal patterns of lake drainage across northern Alaska [6,17,18,34,35,37,45–47]. In addition to landscape history, interacting climate and environmental controls are likely responsible for elevated rates of lake drainage, as anomalous precipitation, temperature, and snow cover may influence patterns of lake change [17,18,37,45]. In the yedoma-dominated tundra of the northern Seward Peninsula, abrupt lake drainage triggered by the gradual expansion of lake boundaries [6] is also hastened by increases in precipitation that facilitate bank overtopping and gully formation [35]. More recently, warmer and wetter climates have triggered potential widespread talik formation, likely responsible for the recent pulse of lake drainage in northwestern Alaska [17,18]. Intensive investigation of decadal patterns and drivers of lake drainage was beyond the scope of this analysis, as such processes have been widely documented and explored. Instead, we focused on the potential resiliency of high-latitude lakes to the suite of geomorphological and climatological processes that can result in drainage and thus instead examined the linkage between spatiotemporal patterns of lake drainage and lake origin derived by the TSI index.

Using morphological observations of lake thaw settlement and roundness, the TSI index differentiates thermokarst and non-thermokarst lakes but requires ancillary datasets to separate lakes into lake origin classes (i.e., Thermokarst, Yedoma, Glacial, and Maar lakes; Figure 6). With the assumption that Thermokarst (non-yedoma) lakes are generally characterized by low topographic relief between the lake bluff and the lake water [8], the TSI improves the remote detection of thermokarst and non-thermokarst lakes within topographically heterogeneous terrain. However, misclassifications were detected. The majority of known misclassifications were identified at the boundary of recently drained lakes, where the initial lake area contracted into smaller lakes within the initial lake bound-

ary. The boundary of these small secondary or tertiary lakes was misinterpreted by the TSI as having limited thaw settlement, thus misclassifying Yedoma or Glacial lakes as non-yedoma Thermokarst lakes. Because most of our study region was located in upland tussock-shrub tundra, these misclassifications were minimal. However, future lake origin mapping efforts will be improved by including the recently mapped distribution of drained thaw lake basins [48] for estimating the probability of thermokarst versus non-thermokarst lakes [10,40,49]. Using this approach, first-order wall-to-wall lake origin maps of Alaska and the circumpolar Arctic may be possible, but uncertainties will be dependent on knowledge of the spatial distribution of yedoma deposits and glacial history. Nevertheless, even in the absence of ancillary datasets, the TSI will provide good baseline information of the distribution of thermokarst and non-thermokarst lakes, while providing lake-specific geomorphologic predisposition of periglacial lakes to thermokarst (lake expansion or drainage) and/or thermo-erosion (retrogressive thaw slumps).

As the climate of the Arctic and subarctic becomes warmer and wetter, we should expect landscape evolution associated with lake dynamics to disproportionately impact biogeophysical processes across permafrost terrains [18,35,50,51]. Ice-rich yedoma and non-yedoma thermokarst lakes will likely experience greater rates of lake expansion and drainage than glacial and Maar lakes underlain by ice-poor permafrost and/or bedrock (Table 2, [7,8]). Although some glacial and yedoma lakes may be relatively stable, due to their steep lake bluffs (associated with deep thaw settlement), they are more vulnerable to climate-triggered mass wasting events such as retrogressive thaw slumps and debris-flows [52,53], which deposit fibrous organic soil mats and mineral sediment in the lake water. These mass-wasting events are detectable via lake water discoloration using remote sensing [37,54], where the rate of deposition (new substrate inputs) into the lake water column is linked with elevated methane emissions [55]. As such, increased labile nutrient deposits in yedoma lakes have resulted in some of the highest point-source methane emissions [56,57], but regional land-atmosphere fluxes of methane have been greater in glacial lakes due to their larger area extent [58]. Thus, improving the representation of landscape heterogeneity may not only reduce uncertainties in tundra carbon dynamics [59] but also identify regions most vulnerable to thermokarst-mediated landscape evolution (Table 2, [17,18]).

## 5. Conclusions

The processes controlling lake formation, expansion, and drainage are likely to vary with lake origin, ground ice content, and the amplitude of climate forcing. Due to the disproportionately large amount of carbon stored in permafrost soils (~50% of the global organic carbon pool) and specifically in yedoma permafrost [60], improving knowledge of the factors controlling periglacial lake drainage is necessary to reduce uncertainties in projected carbon-climate feedbacks. The Thermokarst lake Settlement Index (TSI) will improve the representation of vulnerable versus resilient Arctic and subarctic lakes and provide baseline information in which Arctic hydrologists, ecologists, and geomorphologists may evaluate the likelihood of lake-specific thermokarst dynamics across variable physiography and environmental gradients in permafrost ecosystems.

**Author Contributions:** Conceptualization, methodology, formal analysis, and original draft preparation, M.J.L.; review and editing and funding acquisition, M.J.L. and M.L.C. All authors have read and agreed to the published version of the manuscript.

**Funding:** This research was supported by the National Science Foundation's EnvE-Environmental Engineering program (grant Nos. 1928048 and 1927772).

**Institutional Review Board Statement:** Not applicable.

**Informed Consent Statement:** Not applicable.

**Data Availability Statement:** All data (code and assets/shapefiles) generated in this study can be viewed and downloaded within Google Earth Engine (https://code.earthengine.google.com/?accept_repo=users/mjlara71/TSI; accessed on 1 October 2020) or clone Git repository (git clone https://earthengine.googlesource.com/users/mjlara71/TSI; accessed on 1 October 20). The uploaded asset (https://code.earthengine.google.com/?asset=users/mjlara71/LakeOrigin; accessed on 1 October 2020) includes the following columns: LakeID (identifier), Area_85_89 (lake area during time-period 1985–1989 in ha), Roundness (estimated during 1985–1989), Perimeter (estimated during 1985–1989 in m), Area_15_19 (lake area during time-period 2015–2019 in ha), ElevLake (estimated lake-level elevation in m), ElevBuff (estimated adjacent terrain elevation in m), TSI (Thermokarst lake Settlement Index), LakeChg (percent change in lake area), LakeSettle (estimated lake settlement in m). The late Cenozoic volcanic centers map, database of Ice-rich Yedoma Permafrost, and the Pleistocene Maximum Wisconsinan glacier extent are found at [29,42,43], respectively. All maps and products use the Geographic Coordinate System and Projection: GCS North American 1983 and NAD 1983 Alaska Albers.

**Acknowledgments:** We are grateful for the geospatial support provided by the Polar Geospatial Center. Any use of trade, firm, or product names is for descriptive purposes only and does not imply endorsement by the U.S. Government.

**Conflicts of Interest:** The authors declare no conflict of interest.

## Appendix A

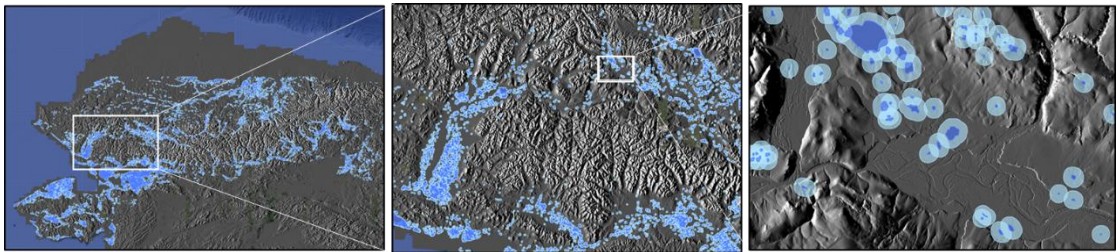

**Figure A1.** Lake-specific 500 m buffers used with the Arctic DEM to compute lake settlement.

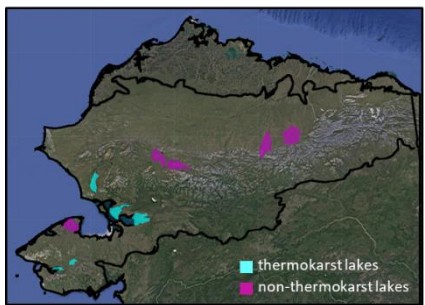

**Figure A2.** Lake sub-regions that correspond with relatively homogenous thermokarst (non-yedoma) and non-thermokarst lakes (including yedoma and glacial lakes).

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
