# Peer review of "Periglacial Lake Origin Influences the Likelihood of Lake Drainage in Northern Alaska"

_remotesensing, doi:10.3390/rs13050852_

Round 1

Reviewer 1 Report

The authors present a discussion about the periglacial lakes and thermokarst lakes settlement index in high latitudes.  It is clear that the authors have assembled a large amount of information and presented a clarify methodology. The manuscript is well organized and and too make a useful contribution for permafrost regions investigations. I recommend the publication in Remote Sensing with minor revision.  The comments are included above:

-The Figures 1, Figure 6 and S1 S2: need Coordinate Reference System (CRS), and Map projection information in maps.

-131 to 143 lines are copies of the 118 to 130 lines.

Author Response

The authors present a discussion about the periglacial lakes and thermokarst lakes settlement index in high latitudes.  It is clear that the authors have assembled a large amount of information and presented a clarify methodology. The manuscript is well organized and and too make a useful contribution for permafrost regions investigations. I recommend the publication in Remote Sensing with minor revision.  The comments are included above:

-The Figures 1, Figure 6 and S1 S2: need Coordinate Reference System (CRS), and Map projection information in maps.

Thanks for your review. All data are provided within the data availability statement and data citations. Users are able to project and reproject as needed. To keep figure captions as short as possible, we elected to forego adding CRS and projections.

-131 to 143 lines are copies of the 118 to 130 lines.

We deleted this section.

Reviewer 2 Report

The paper introduces a method to use remote sensing data (Landsat images) to determine the origin of Arctic and sub-Arctic lakes in Alaska and examines whether the origin of periglacial lakes can be linked to their likelihood of drainage. This is done by producing maps of lake origin from 1985 and 2019 and examining the pattern of lake drainage.
The authors define a Thermokarst lake Settlement Index (TSI) based on data from Landsat images and elevation data from the Arctic Digital Elevation Model which is a continuous index ranging from negative (thermokarst lakes) to positive (non-thermokarst) values. By means of ancillary data sets (United States Geological Survey map of late Cenozoic volcanic centers, database of Ice-rich Yedoma Permafrost, Pleistocene Maximum Wisconsinan glacier extent). Based on TSI values and the ancillary data, lakes were classified as either a)non-Yedoma thermokarst lakes, b)Yedoma thermokarst lakes, c)Glacial lakes and d)Volcanic (Maar) lakes.
Two separate maps of lake classification are produced; one for 1985-1989 and one for 2015-2019 and these maps are used to investigate dependence of lake drainage on lake origin, determining different likelihoods of drainage depending on lake type, with thermokarst lakes having the highest drainage probability.
The paper shows an interesting method to pinpoint the lakes most vulnerable to drainage in a given area, which is interesting in a climate change perspective. The central element in the method is remote sensing data (landsat images), however, a good deal of ancillary information is needed as well (United States Geological Survey map of late Cenozoic volcanic centers, database of Ice-rich Yedoma Permafrost, Pleistocene Maximum Wisconsinan glacier extent) and is, as such, only fully applicable in areas where some degree of mapping has already taken place.

Minor points: The section Image processing (line 118-138) is repeated in lines 131-143.

Author Response

The paper introduces a method to use remote sensing data (Landsat images) to determine the origin of Arctic and sub-Arctic lakes in Alaska and examines whether the origin of periglacial lakes can be linked to their likelihood of drainage. This is done by producing maps of lake origin from 1985 and 2019 and examining the pattern of lake drainage.
The authors define a Thermokarst lake Settlement Index (TSI) based on data from Landsat images and elevation data from the Arctic Digital Elevation Model which is a continuous index ranging from negative (thermokarst lakes) to positive (non-thermokarst) values. By means of ancillary data sets (United States Geological Survey map of late Cenozoic volcanic centers, database of Ice-rich Yedoma Permafrost, Pleistocene Maximum Wisconsinan glacier extent). Based on TSI values and the ancillary data, lakes were classified as either a)non-Yedoma thermokarst lakes, b)Yedoma thermokarst lakes, c)Glacial lakes and d)Volcanic (Maar) lakes. Two separate maps of lake classification are produced; one for 1985-1989 and one for 2015-2019 and these maps are used to investigate dependence of lake drainage on lake origin, determining different likelihoods of drainage depending on lake type, with thermokarst lakes having the highest drainage probability.
The paper shows an interesting method to pinpoint the lakes most vulnerable to drainage in a given area, which is interesting in a climate change perspective. The central element in the method is remote sensing data (landsat images), however, a good deal of ancillary information is needed as well (United States Geological Survey map of late Cenozoic volcanic centers, database of Ice-rich Yedoma Permafrost, Pleistocene Maximum Wisconsinan glacier extent) and is, as such, only fully applicable in areas where some degree of mapping has already taken place.

Thanks for your review of our article. You are correct, a key limitation of the formulation of the Lake origin map was in the availability of ancillary data layers such as those referenced. We highlight these limitations on lines 300-302 & 310-314, and further detail the usefulness of the TSI as a stand alone index on lines 302-309 & 315-318. Furthermore, the key data layers we used to separate glacial, yedoma, and thermokarst lakes are currently available across the Arctic (line 313), though we are unaware of a pan-Arctic volcanic origin dataset.

Minor points: The section Image processing (line 118-138) is repeated in lines 131-143.

We deleted this section.

Reviewer 3 Report

Overall an excellent research article. Clearly written, well presented, straight-forward.

-- While Fig. 1 is an appropriate and useful figure, I wouldn't necessarily assume from the outset the all potential readers are familiar with the geography of Alaska or the broader Arctic and can therefore easily place the study area. Perhaps another figure should be added or the current figure modified in some way (e.g., another inset) to situate the study area within the whole of Alaska or better yet within the broader region (e.g., including parts of Canada and Russia), or something similar. As it presently stands, only someone already very familiar with Alaska could quickly understand your study area and how it fits into the bigger high latitude picture. 

-- I'm sure this would get corrected during copy editing later, but will point out anyway: Line 315 transitions from Palatino into Arial, and then switches back again mid-sentence on line 316. 

Author Response

Overall an excellent research article. Clearly written, well presented, straight-forward.

Thank you

-- While Fig. 1 is an appropriate and useful figure, I wouldn't necessarily assume from the outset the all potential readers are familiar with the geography of Alaska or the broader Arctic and can therefore easily place the study area. Perhaps another figure should be added or the current figure modified in some way (e.g., another inset) to situate the study area within the whole of Alaska or better yet within the broader region (e.g., including parts of Canada and Russia), or something similar. As it presently stands, only someone already very familiar with Alaska could quickly understand your study area and how it fits into the bigger high latitude picture. 

Great point, we added another map inset representing our study region within pan-Arctic spatial domain.

-- I'm sure this would get corrected during copy editing later, but will point out anyway: Line 315 transitions from Palatino into Arial, and then switches back again mid-sentence on line 316. 

We deleted this section and corrected the font.